# Effect of Deformation Conditions on Strain-Induced Precipitation of 7Mo Super-Austenitic Stainless Steel

**DOI:** 10.3390/ma16196401

**Published:** 2023-09-25

**Authors:** Shiguang Xu, Jinshan He, Runze Zhang, Fucheng Zhang, Xitao Wang

**Affiliations:** 1Collaborative Innovation Center of Steel Technology, University of Science and Technology Beijing, Beijing 100083, China; xushiguang974@163.com (S.X.); b20190521@xs.ustb.edu.cn (R.Z.); 2State Key Laboratory of Metastable Materials Science and Technology, Yanshan University, Qinhuangdao 066004, China; zfc@ysu.edu.cn; 3Shandong Provincial Key Laboratory for High Strength Lightweight Metallic Materials, Advanced Materials Institute, Qilu University of Technology (Shandong Academy of Science), Jinan 250353, China

**Keywords:** 7Mo super-austenitic stainless steel, strain-induced precipitation, stress relaxation test, deformation parameters

## Abstract

Strain-induced precipitation (SIP) behaviors of 7Mo super-austenitic stainless steel (SASS) under various deformation conditions were studied by stress relaxation tests. The research demonstrates that sigma phases are the primary SIP phases of 7Mo SASS. Generally, SIP is mainly distributed in granular shape at the boundaries of deformed grains or recrystallized grains, as well as around the deformed microstructure, such as deformation twin layers/matrix interfaces. The variation of deformation parameters can lead to changes in microstructure, therefore influencing the distribution of SIP. For instance, with the temperature increases, the SIP distribution gradually evolves from deformed grain boundaries to recrystallized grain boundaries. The average size of SIP increases with increasing temperature and strain, as well as decreasing strain rate. The SIP content also increases with increasing strain and decreasing strain rate, while exhibiting an initial rise followed by a decline with increasing temperature, reaching its maximum value at 850 °C. The presence of SIP can promote recrystallization by particle-induced nucleation (PSN) mechanism during the hot deformation process. Moreover, the boundaries of these recrystallized grains can also serve as nucleation sites for SIP, therefore promoting SIP. This process can be simplified as $SIP\overset{PSN}{\to }Recrystallization\overset{Nucleation\;sites}{\to }SIP$. With the increase in holding time and the consumption of stored energy, the process gradually slows down, leading to the formation of a multi-layer structure, namely SIPs/Recrystallized grains/SIPs structure. Moreover, SIP at recrystallized grain boundaries can hinder the growth of recrystallized grains. Through this study, a comprehensive understanding of the SIP behaviors in 7Mo SASS under different deformation conditions has been achieved, as well as the interaction between SIP and recrystallization. This finding provides valuable insights for effective control or regulation of SIP and optimizing the hot working processes of 7Mo SASS.

## 1. Introduction

Super-austenitic stainless steel (SASS) has exceptional corrosion resistance and favorable mechanical properties due to the extremely high alloying elements, such as Cr, Mo, and N, rendering it widely utilized in harsh environments, such as petrochemical and seawater desalination industries [1]. However, the extremely high content of alloying elements also greatly facilitates the formation of second phases, such as the sigma phase, during solidification [2] or aging [3,4]. In addition, the presence of deformation can significantly further promote the precipitation of second phases [1,5,6,7]. This can be attributed to the abundant lattice defects, such as dislocations and microbands, generated by plastic deformation, which can serve as nucleation sites and provide rapid diffusion channels for precipitation [8]. The phenomenon of rapid precipitation resulting from deformation is commonly referred to as strain-induced precipitation (SIP) [9].

Previous studies on SIP have primarily focused on microalloyed steels [10,11,12], Al-Mn alloy [13], and nickel-based alloys [14,15], encompassing the precipitation behaviors [8] and kinetics models [13,14,15,16]. They found that in microalloyed steels, SIP was mainly carbide, nitride, or carbonitride precipitates [17,18,19], which exhibited extremely small sizes at the nanoscale and were mainly distributed in dislocations [20]. When in a nickel-based alloy, SIP is mainly M_23_C_6_ [14], which initially nucleates in high-density dislocations regions inside grains and subsequently at grain boundaries [15]. Moreover, several variables, including temperature, strain, and strain rate, exhibit significant influence on SIP behaviors. For instance, An et al. [19] found that in the transient deformation stage, the volume density of MnS increased and the diameter decreased with increasing strain. When in the steady deformation stage, the volume density and diameter of MnS remain nearly unchanging with increasing strain. The presence of SIP can affect the flow behaviors [14] and recrystallization [12,21]. For instance, static recrystallization can be significantly hindered by SIP at the initial stage, while it continues to occur after SIP is finished [20]. Moreover, SIP with different sizes shows various effects of recrystallization. SIP with a small size can hinder recrystallization, whereas second phases with a coarse size can facilitate recrystallization [22]. However, there is limited research on the SIP behaviors of SASS. Pu et al. observed many strain-induced sigma phases in 6Mo [6] and 7Mo SASS [23] after hot deformation. However, no further investigations have been conducted on SIP behaviors. SIP in SASS is mainly a brittle intermetallic phase, such as the sigma phase. The presence of many brittle intermetallic phases promotes deformation cracking [6], therefore increasing the processing difficulty and reducing the corrosion resistance of SASS. Therefore, a systematic investigation of SIP behaviors of SASS is essential to optimize the hot working process and microstructure for SASS.

The purpose of this study is to investigate the recrystallization and SIP behaviors of 7Mo SASS under various deformation conditions through stress relaxation tests, including the identification and distribution of SIP, the variation of SIP size and content, and the relationship between SIP and recrystallization.

## 2. Materials and Methods

The as-cast 7Mo SASS ingots were supplied by Taiyuan Iron and Steel Company (TISCO, Taiyuan, China), with a chemical composition (wt.%) of 0.01C, 0.02Si, 5.3Mn, 24.7Cr, 19.6Ni, 7.1Mo, 0.5N, 0.4Cu, 0.001Ce and balance Fe. A steel ingot with dimensions of 60 × 50 × 40 mm was cut from the as-cast 7Mo SASS ingots. Subsequently, the steel ingot was first hot-forged at 1200 °C with 50% reduction (from 60 mm to 30 mm), followed by the solution being treated at 1250 °C for 3 h, and finally quenched in water to obtain compression specimens. Figure 1a shows a schematic diagram of the specimen preparation process. Stress relaxation tests were conducted on Gleeble-3500D (Dynamic Systems Inc., Atlanta, GA, USA), using cylindrical compression specimens of Φ 8 × 12 mm. The selected experimental parameters for stress relaxation tests are as follows: the temperature of 750–1100 °C, the strain rate of 0.01–1 s^−1^, the strain of 20–60% (the true strain of 0.22–0.92), and the holding time of 1000 s. Due to the strong precipitation tendency of 7Mo SASS, specimens after the stress relaxation test were quickly quenched in water to fix the microstructure. Figure 1b shows the schematic diagram of the stress relaxation tests used in this study.

Specimens for X-ray diffraction (XRD, Ultima IV, Rigaku, Tokyo, Japan), electron probe X-ray micro-analyzer (EPMA, EPMA-8050G, SHIMADZU, Tokyo, Japan), electron channeling contrast imaging (ECCI, Gemini450, Zeiss, Oberkochen, Germany) and electron backscatter diffraction (EBSD, Gemini450, Zeiss, Oberkochen, Germany) were ground from 60 to 2000 grit, followed by polished from 2.5 to 0.5 μm, and finally subjected to electropolishing using a mixture of 15 vol.% perchloric acid and 85 vol.% ethyl alcohol for 15 s. The voltage and current for electropolishing were 15 V and 1 A, respectively. The XRD measurement conditions were as follows: 2θ of 20~90°, step size of 0.02° and scan rate of 5°/min. The voltage and current were set at 20 kV and 10 nA for EBSD, respectively, and were set at 25 kV and 2 nA for ECCI. The step size of EBSD was set at 0.8 μm. The EBSD data were analyzed by AZtecCrystal software (2.0). More than 10 ECC images with an area of 1380 μm × 930 μm were used for statistical analysis. Foils for transmission electron microscope (TEM, Tecnai G2 20, FEI, Hillsboro, OR, USA) were prepared from the disks with a twin-jet polisher at 50 mV in an electrolyte consisting of 92 vol.% ethanol and 8 vol.% perchloric acid at −25 °C.

## 3. Results

### 3.1. Initial Microstructure

Figure 2 shows the initial microstructure and the composition distribution of 7Mo SASS before stress relaxation tests. ECC image in Figure 2a and XRD results in Figure 2b show that the initial microstructure is composed of austenitic matrix (γ-Fe) and undissolved sigma phases at grain boundaries. EDS results in Figure 2a indicate that undissolved sigma phases are rich in Cr and Mo, with specific contents of 29.2 wt.% and 12.0 wt.%, respectively, which are 3.3 wt.% and 4.5 wt.% higher than those of the matrix. According to statistical analysis, the average grain size of γ-Fe is 135 μm. In addition, the content and average size of undissolved sigma phases are 3.8 vol.% and 23.4 μm, respectively. EPMA maps in Figure 2c indicate that the distribution of elements in γ-Fe is relatively uniform, and no obvious element segregation at grain boundaries.

### 3.2. Deformation Microstructure and SIP Behaviors at Different Temperature

Figure 3 and Figure 4 show the inverse pole figure (IPF) and kernel average misorientation (KAM) images of 7Mo SASS after stress relaxation tests at 750–1100 °C, respectively. Generally, the microstructure and recrystallization fraction exhibit significant variations at different temperatures. At 750–850 °C, as shown in Figure 3a–c, the microstructure is mainly composed of deformed grains with a low recrystallization fraction (about 10%, as shown in Figure 5). Further observation reveals the presence of obvious deformation twins and microbands inside deformed grains. KAM images in Figure 4a–c reveal that there is a high strain concentration within the deformed microstructure, which is mainly distributed at the boundaries of deformed grains and microbands, as well as around undissolved sigma phases. When the temperature reaches 900 °C, as shown in Figure 3d, the microstructure remains predominantly composed of deformed grains, while the recrystallization fraction slightly increases to 16.9% (Figure 5). The magnified IPF image in Figure 3d reveals the presence of obvious small, recrystallized grains at grain boundaries and some coarse recrystallized grains inside deformed grains. In addition, the KAM image in Figure 4d shows many obvious low KAM regions at deformed grain boundaries, which is consistent with the distribution of recrystallization. The presence of recrystallization leads to a decrease in KAM value, from 1.44° to 1.10°. When the temperature further increases to 1000 °C, as shown in Figure 3e, the recrystallization fraction shows a significant increase to 97.9% (Figure 5), and recrystallized grains play a dominant role in the microstructure. Such high recrystallization fraction also leads to a significant reduction in KAM value, with an average KAM value of 0.18°, representing only 16.4% of that at 900 °C. This phenomenon also indicates that the extent of recrystallization of 7Mo SASS is greatly affected by temperature. Further observation of the IPF image in Figure 3e reveals the presence of fine recrystallized grain regions around undissolved sigma phases. The KAM image in Figure 4e shows obvious strain concentration in some recrystallized grains around undissolved sigma phases. This phenomenon indicates that large strain tends to accumulate in recrystallized grains surrounding undissolved sigma phases during the deformation process, leading to the transformation of these recrystallized grains into deformed grains. Subsequently, during the holding process, further recrystallization will take place in this region, resulting in the appearance of fine recrystallized grain regions. When the temperature reaches 1100 °C, as shown in Figure 3f, the microstructure is mainly equiaxed recrystallized grains with a large average grain size of 99.6 μm (Figure 5). This observation indicates the substantial growth of recrystallized grains during the holding process.

Figure 6 shows the ECC images of 7Mo SASS after stress relaxation tests at 750–1100 °C. At 750–850 °C, as shown in Figure 6a–c, strain-induced precipitates (SIPs) are mainly distributed in granular shape at deformed grain boundaries, as well as a small number of granular SIPs are also observed inside deformed grains. At 850 °C, as shown in Figure 6c, obvious needle-like SIPs are also observed inside deformed grains. At 900 °C, as shown in Figure 6d, in addition to the deformed grain boundaries, SIPs are also observed at recrystallized grain boundaries. Notably, these SIPs can envelop the recrystallized grains (Figure 6(d1,d2)). When the temperature increases to 1000 °C, as shown in Figure 6e, SIPs are distributed in granular shape along recrystallized grain boundaries, while exhibiting a bimodal size distribution. In addition, as shown in Figure 6e, distinct sub-grains are observed within some recrystallized grains. These sub-grains can further develop into recrystallized grains with the increase in strain or holding time, therefore leading to a refinement of grain structure. This finding is consistent with the results of the KAM image in Figure 4e. When the temperature further increases to 1100 °C, as shown in Figure 6f, no obvious SIPs are observed. This phenomenon may be attributed to the significant decrease in deformation stored energy (Figure 4) resulting from temperature elevation. Consequently, sigma phases are difficult to precipitate or undergo dissolution during the holding process.

Figure 7 shows the XRD results and the statistical data of SIP after the stress relaxation test under different temperatures. The XRD results in Figure 7a show that the sigma phase is the primary SIP under experimental temperature. Moreover, the most significant XRD peaks of the sigma phase are observed at 850 °C, indicating that the content of SIP reaches its maximum value at this temperature. The statistical data in Figure 7b shows that the average size of SIP continues to increase with increasing temperature, reaching its maximum value of 668.4 nm at 1000 °C. This can be attributed to the significant enhancement in diffusion rate with increasing temperature. In microalloyed steels, the average size of SIP (carbide, nitride or carbonitride precipitates) is generally 2~20 nm [12,21,24]. As shown in Figure 7b, the average size of SIP (sigma phases) in 7Mo SASS is about 159.2~668.4 nm, which is about 8~335 times larger than that in microalloyed steels. The two primary reasons are as follows: First, 7Mo SASS contains high levels of sigma-forming elements, such as the Mo content is 7.1 wt.% in this study. The trace elements that contribute to SIP formation in microalloyed steels are significantly low, such as Nb content typically ranging from approximately 0.01 to 0.2 wt.% [18,21,25]. Moreover, sigma-forming elements have a small atomic size compared to trace elements. For instance, the atomic size of Mo and Nb are 1.40 Å and 1.46 Å, respectively. Small atomic size indicates a fast diffusion rate. The content of SIP first increases and then decreases with increasing temperature, reaching its maximum value of 4.3 vol.% at 850 °C. This observation is consistent with the XRD results in Figure 7a. The content of SIP is determined by its quantity and size. Although the average size of SIP continues to increase with increasing temperature, the quantity of SIP significantly decreases due to the large reduction of stored energy, as shown by the KAM value in Figure 5. Ultimately, this leads to the highest SIP content at 850 °C.

### 3.3. Deformation Microstructure and SIP Behaviors at Different Strain Rate

Figure 8 shows the IPF and KAM images of 7Mo SASS after stress relaxation tests at 0.01–1 s^−1^. IPF images in Figure 8 reveal that the microstructure at different strain rates predominantly consists of deformed grains. As shown in Figure 9, the recrystallization fraction and average recrystallized grain size exhibit slight variations at different strain rates. This phenomenon indicates that strain rate has a limited effect on the recrystallization of 7Mo SASS. However, KAM images in Figure 8 indicate that with the decrease of strain rate from 1 s^−1^ to 0.01 s^−1^, the average KAM value continuously decreases from 1.48° to 1.12°. The decrease in KAM value can be attributed to the substantial increase in deformation time (from 0.69 s to 69 s) caused by a lower strain rate, therefore increasing the extent of dynamic recovery. In addition, at 0.01 s^−1^, as shown in Figure 8a, obvious high-angle grain boundaries (HAGBs) can be observed in deformed grains. These HAGBs are developed by the continuous absorption of surrounding dislocations by high-density dislocation walls (DDWs), commonly referred to as recovery. The presence of low KAM regions surrounding these HAGBs provides supporting evidence, as shown in Figure 8b. With the strain rate increasing to 0.1 s^−1^ or 1 s^−1^, as shown in Figure 8c–f, no obvious low KAM regions are observed in deformed grains, while high-density deformation twins and microbands are evident.

Figure 10 shows the ECC images of 7Mo SASS after stress relaxation tests at 0.01–1 s^−1^. ECC images in Figure 10 show that SIPs are distributed in granular shape at grain boundaries or inside deformed grains. In addition, a small number of needle-like SIPs are observed inside deformed grains. As shown in Figure 10d, with the decrease in strain rate, there is a continuous increase observed in both the content and average size of SIP, ranging from 3.9 vol.% to 5.6 vol.% and from 187.2 nm to 412.6 nm, respectively. The decrease in strain rate represents the increase in deformation time, such as the deformation time ranging from 0.69 s to 69 s with the strain rate decreasing from 1 s^−1^ to 0.01 s^−1^. Our previous research [26] reveals that for 7Mo SASS, the strain-induced sigma phase can be observed at the initial stage of deformation (earlier than 20% strain). Therefore, the elongation in deformation time signifies an increase in SIP time, which leads to an increase in SIP size. Due to the limited decrease in stored energy with a decreasing strain rate, as shown by the KAM value in Figure 8, there is only a slight decrease in the quantity of SIP. Consequently, the variation of SIP content is predominantly governed by its size change, therefore demonstrating a consistent upward trend as the strain rate decreases.

### 3.4. Deformation Microstructure and SIP Behaviors at Different Strain

Figure 11 and Figure 12 show the IPF and KAM images of 7Mo SASS after stress relaxation tests at 20–60% strain. IPF images in Figure 11 reveal that the microstructure under different strains is mainly composed of deformed grains. With the increase in strain, as shown in Figure 11f, the recrystallization fraction shows a continuous increase from 0.5% to 11.7%, while the average recrystallized grain size shows minor fluctuations. Further observation reveals that when the strain ranges from 20% to 30%, as shown in Figure 11a,b and Figure 12a,b, distinct microbands can be observed inside deformed grains. This phenomenon suggests that deformation is controlled by microband-induced plasticity (MBIP) at the initial deformation stage. When the strain increases to 40%, as shown in Figure 11c and Figure 12c, a small number of deformation twins are observed inside deformed grains, indicating the occurrence of twinning-induced plasticity (TWIP). With the strain further increases to 50% and 60%, as shown in Figure 11d,e and Figure 12d,e, there is a significant increment in the content of deformation twins, indicating a substantial enhancement of TWIP. The formation of deformation twins can be attributed to the formation of intense local stress concentration with increasing strain, especially at deformed grain boundaries, as shown in KAM images in Figure 12. Therefore, deformation twins mainly propagate from grain boundaries into the interior of grains, as shown in Figure 11c–e. KAM images reveal that strain concentration mainly occurs at the boundaries of deformed grains and microbands, as well as around undissolved sigma phases. With the increase in strain, both the level of strain concentration and the average KAM value continue to increase.

Figure 13 shows the ECC images of 7Mo SASS after stress relaxation tests at 20–60% strain. At 20–30% strain, as shown in Figure 13a,b, SIPs are mainly continuously distributed in granular shape at deformed grain boundaries, and they virtually exist at all deformed grain boundaries. With the further increase in strain, as shown in Figure 13c–e, obvious SIPs can be observed inside deformed grains, and their content increases with increasing strain. The statistical data in Figure 13f reveal that both the content and the average size of SIP increase with increasing strain. The increase in SIP size can be attributed to the slight increase in deformation time (from 2.2 s to 9.2 s) and the significant increase in stored energy (from 0.73° to 1.56°). The presence of large stored energy indicates the existence of high-density dislocations and vacancies [27], which not only facilitates the nucleation of SIP but also enhances the diffusion of elements, therefore promoting SIP growth. When the strain exceeds 30%, there is a rapid increase in SIP content, which can be attributed to the large occurrence of SIP inside deformed grains. The appearance of intragranular SIP is a result of the formation of deformed structures during extensive deformation, such as deformation twins or microbands, which provide nucleation sites for SIP. Moreover, large stored energy also facilitates intragranular SIP nucleation. When the strain exceeds 40%, the significantly increased deformation stored energy (Figure 12) leads to an accumulation of high-density dislocations and vacancies, therefore promoting SIP growth and causing a rapid increase in SIP size.

## 4. Discussion

### 4.1. The Effect of Deformation Parameters on Recrystallization Behaviors

Figure 14 shows the distribution of recrystallization, average grain size, and average KAM value of 7Mo SASS under different deformation conditions. Generally, the recrystallization fraction exhibits an increasing trend with rising temperature and strain, while showing minimal correlation with strain rate. The order of the influence of deformation parameters on recrystallization fraction can be ranked as follows, from strong to weak: temperature, strain, and strain rate. The elevated temperature effectively enhances the mobility of grain boundaries, therefore promoting the growth of recrystallized grains and increasing the recrystallization fraction. For instance, when the temperature increases from 900 °C to 1000 °C, the recrystallization fraction increases from 16.9% to 97.9%. With the temperature further increasing to 1100 °C, complete recrystallization occurs, and the substantial growth of recrystallized grains is observed, as shown in Figure 14b. The occurrence of substantial recrystallized grains significantly consumes deformation stored energy, leading to a notable decrease in KAM value, as shown in Figure 14c. The elevated strain can significantly facilitate recrystallization nucleation by generating a large amount of stored energy, as shown in Figure 14c, while exerting limited influence on recrystallization growth, therefore resulting in a slight increment in recrystallization fraction. However, the decrease in strain rate only exerts a limited influence on stored energy by increasing deformation time and recovery extent. Moreover, due to the decrease of stored energy and slow recrystallization rate at low temperatures, there is a limited change in recrystallization fraction despite long deformation time (recrystallization time). Therefore, strain rate shows a negligible effect on recrystallization fraction.

### 4.2. The Effect of Deformation Parameters on SIP Behaviors

According to the XRD (Figure 7a) and TEM (Figure 15) results, SIPs in 7Mo SASS are mainly sigma phases. As shown in ECC images in Figure 13a,b, i.e., at 850 °C with 20–30% strain, SIPs are mainly distributed in the granular shape at deformed grain boundaries. This phenomenon can be attributed to the accumulation of large deformation stored energy at grain boundaries (Figure 12a,b) and the provision of nucleation sites and effective elements diffusion channels. In addition, the granular morphology is advantageous in minimizing the interfacial energy of SIP by reducing the interfacial area, therefore facilitating the nucleation and growth of SIP. With the increase in strain, many deformed structures, including deformation twins, microbands, and DDWs, are generated inside deformed grains, as shown in Figure 12. The deformed structures can also serve as the nucleation sites for SIP, with a large accumulation of stored energy in their vicinity, as shown in KAM images in Figure 12. Therefore, SIP can form around the deformed structures inside deformed grains, and their content increases with increasing strain, as shown in Figure 13. As shown in TEM images in Figure 15a–c, intragranular SIP are mainly distributed as the interface of deformation twin layers and matrix, intragranular HAGBs, and microband boundaries. Further observation indicates that the needle-like SIP is mainly distributed around microbands, and exhibits a distinct orientation relationship with the surrounding matrix, i.e., ($\bar{1}\;\bar{1}\;\bar{1}$)_γ-Fe_//(00$\bar{1}$)_σ_. The existence of an orientation relationship is advantageous in terms of minimizing interfacial energy, and the needle-like shape contributes to the reduction of elastic strain energy, therefore promoting the nucleation and growth of SIP. This orientation relationship has also been reported in other studies [4]. With the increase in temperature, deformed grains are gradually replaced by recrystallized grains, while stored energy and deformed microstructure are rapidly consumed, leading to SIP being mainly distributed at the boundaries of recrystallized grains. However, the distribution of SIP shows limited changes under different strain rates, due to the absence of significant alterations in the deformed microstructure.

TEM images in Figure 15 also reveal the presence of numerous recrystallized grains or sub-grains surrounding SIP. This can be attributed to the accumulation of high-density dislocations around the large SIP during the deformation process, which facilitates recrystallization in its vicinity, namely the particle-induced nucleation (PSN) mechanism [28]. Furthermore, as shown in Figure 6d and Figure 15d, the boundaries of recrystallized grains can also serve as the nucleation sites for SIP, therefore promoting SIP. This process can be simplified as $SIP\overset{PSN}{\to }Recrystallization\overset{Nucleation\;sites}{\to }SIP$. As the increase in holding time, a multi-layered structure of SIP/Recrystallization/SIP is gradually formed, as shown in Figure 15d. Meanwhile, there is a continuous decrease of stored energy with increasing holding time. When the stored energy reaches a low level, recrystallization through PSN becomes difficult, therefore this process eventually ceases. Moreover, the presence of multi-layered structures can seriously inhibit the growth of recrystallized grains among them.

Figure 16 shows the average size and content of SIP in 7Mo SASS under different deformation conditions. Generally, the average SIP size increases with increasing temperature and strain, as well as decreasing strain rate. The order of the influence of deformation parameters on average SIP size can be ranked as follows, from strong to weak: temperature, strain rate, and strain. The average size of SIP is predominantly governed by the element diffusion rate and diffusion time. The elevated temperature significantly enhances the element diffusion rate, therefore profoundly promoting the growth of SIP. Furthermore, strain rate has limited influence on diffusion rate. The decreased strain rate increases deformation time (from 0.69 s to 69 s), therefore increasing the formation time of SIP and resulting in an increase in average SIP size. The increased strain primarily facilitates the diffusion of elements by the provision of high-density dislocations and vacancies, resulting in a slow increase in SIP size. The SIP content exhibits an increasing trend with the increase in strain and the decrease of strain rate, while it initially rises and subsequently declines with the increase in temperature. The influence of deformation parameters on SIP content shows the order, from strong to weak, followed by strain, strain rate, and temperature. Interestingly, this order exhibits an inverse trend with the average SIP size. This is attributed to the fact that SIP content is influenced not only by its size but also by the quantity of SIP. The nucleation of SIP is primarily associated with the deformation of stored energy and exhibits a positive correlation with increasing stored energy. The increased strain can significantly enhance stored energy and vacancy density, therefore effectively facilitating the nucleation of SIP, and slightly promoting the growth of SIP. Consequently, the content of SIP exhibits a rapid increment with increasing strain. From the previous analysis, the decreased strain rate promotes the growth of SIP by extending the deformation time. The increase in deformation time also facilitates recovery, leading to a higher consumption of deformation stored energy and subsequent suppression of SIP nucleation. Therefore, the influence of strain rate on the SIP content is comparatively lower than that of strain. Although the increased temperature significantly promotes SIP growth, the extensive occurrence of recrystallization excessively consumes deformation stored energy, therefore strongly suppressing the nucleation of SIP and resulting in its maximum content at 850 °C.

## 5. Conclusions

The recrystallization and SIP behaviors of 7Mo SASS under different deformation conditions were studied in this work and the following conclusions can be summed up:

The sigma phase is the primary SIP of 7Mo SASS. Strain-induced sigma phases are mainly distributed in granular shape at deformed grain boundaries and around the deformed structures, such as deformation twin layers/matrix interfaces and microband boundaries. Recrystallized grain boundaries can also serve as nucleation sites for strain-induced sigma phases.Recrystallization fraction is primarily governed by temperature, with a significantly lesser impact from strain and strain rate. The order of the influence of deformation parameters on recrystallization fraction, from strong to weak, is as follows: temperature, strain, and strain rate. The average SIP size increases with increasing temperature and strain, as well as decreasing strain rate. The order of the influence of deformation parameters on SIP size is as follows: temperature has the greatest impact, followed by strain rate, and then strain. The SIP content also increases with increasing strain and decreasing strain rate, while exhibiting an initial increase followed by a decrease with increasing temperature, reaching its maximum value as 850 °C. The order of the influence of deformation parameters on the SIP content, from strong to weak, is as follows: strain, strain rate, and temperature. This order is opposite to that observed for SIP size. This is because the SIP content is influenced not only by its size but also by its quantity. The presence of SIP can facilitate recrystallization by the PSN mechanism. Moreover, the boundaries of these recrystallized grains can also serve as nucleation sites for subsequent SIP, promoting SIP formation. This process can be simplified as $SIP\overset{PSN}{\to }Recrystallization\overset{Nucleation\;sites}{\to }SIP$. With the advancement of the holding process and the consumption of stored energy, the process gradually slows down and even stops, leading to the formation of a multi-layered structure. Moreover, SIP at recrystallized grain boundaries can hinder the growth of recrystallization. Therefore, moderate SIP may be utilized to promote recrystallization and refine recrystallized grains.

## Figures and Tables

**Figure 1 materials-16-06401-f001:**
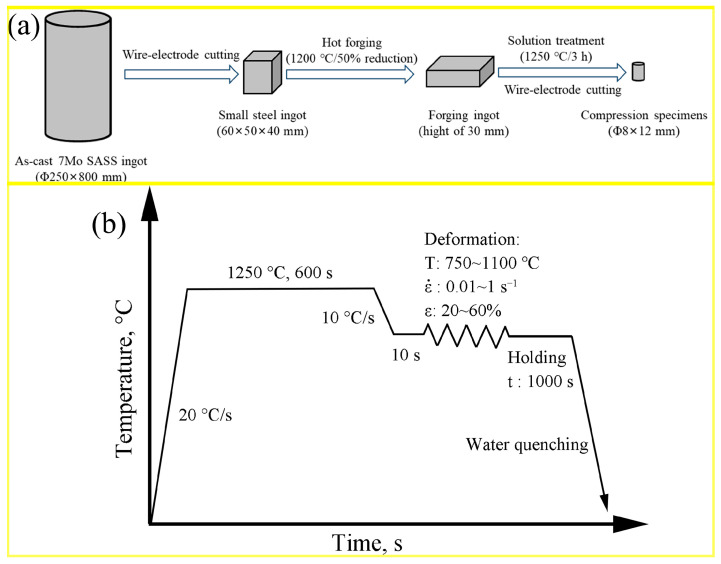
Schematic diagram of (**a**) specimen preparation process and (**b**) stress relaxation tests.

**Figure 2 materials-16-06401-f002:**
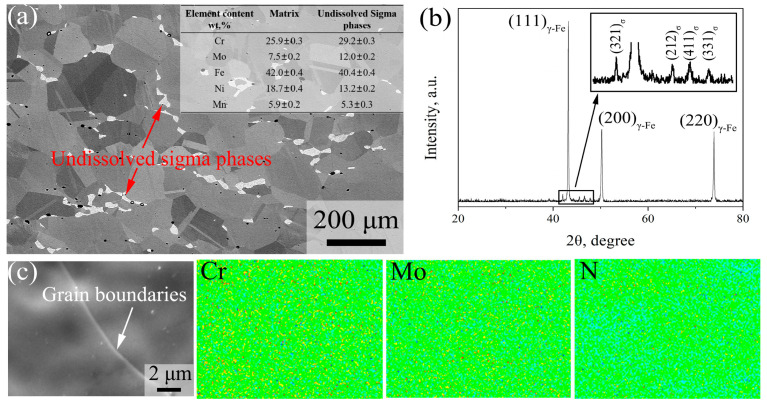
Initial microstructure of 7Mo SASS before stress relaxation test. (**a**) ECC images; (**b**) XRD pattern; (**c**) EPMA maps.

**Figure 3 materials-16-06401-f003:**
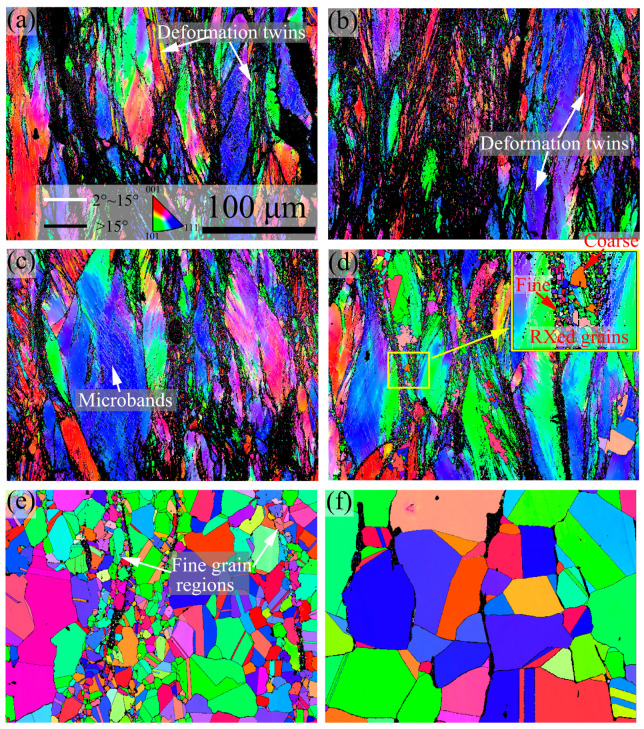
IPF images of 7Mo SASS after stress relaxation tests at 0.1 s^−1^, 50% strain, and 1000 s with different temperatures. (**a**) 750 °C; (**b**) 800 °C; (**c**) 850 °C; (**d**) 900 °C; (**e**) 1000 °C; (**f**) 1100 °C.

**Figure 4 materials-16-06401-f004:**
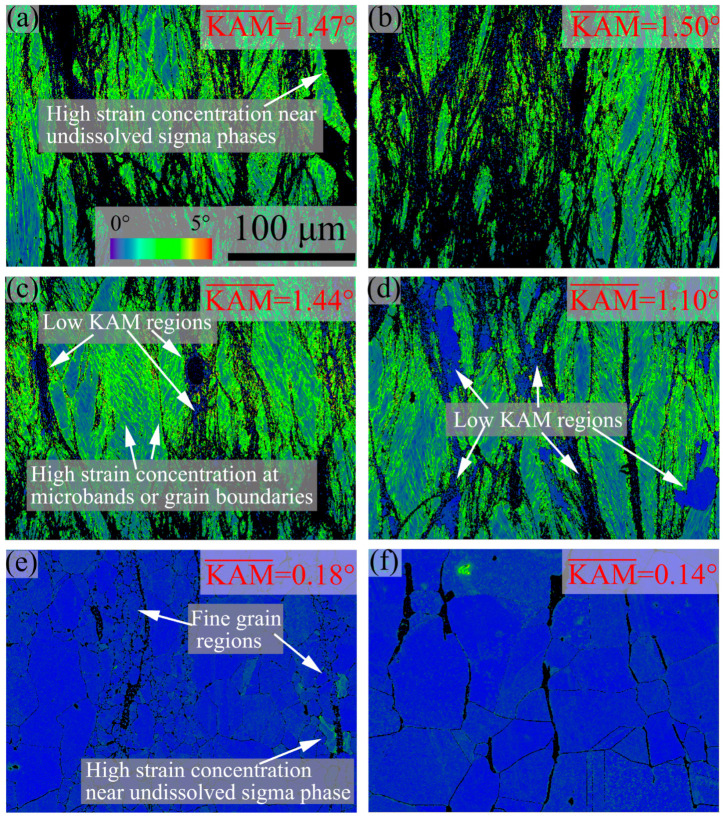
KAM images of 7Mo SASS after stress relaxation tests at 0.1 s^−1^, 50% strain, and 1000 s with different temperatures. (**a**) 750 °C; (**b**) 800 °C; (**c**) 850 °C; (**d**) 900 °C; (**e**) 1000 °C; (**f**) 1100 °C.

**Figure 5 materials-16-06401-f005:**
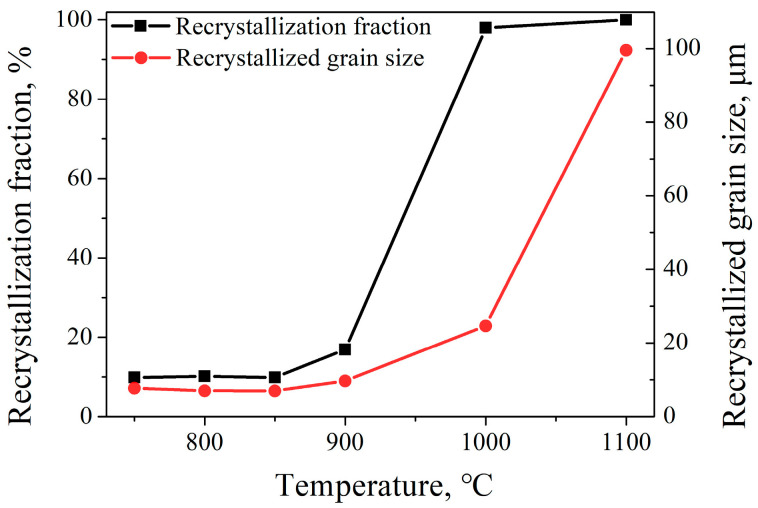
Recrystallization fraction and average recrystallized grain size of 7Mo SASS after stress relaxation tests under different temperatures.

**Figure 6 materials-16-06401-f006:**
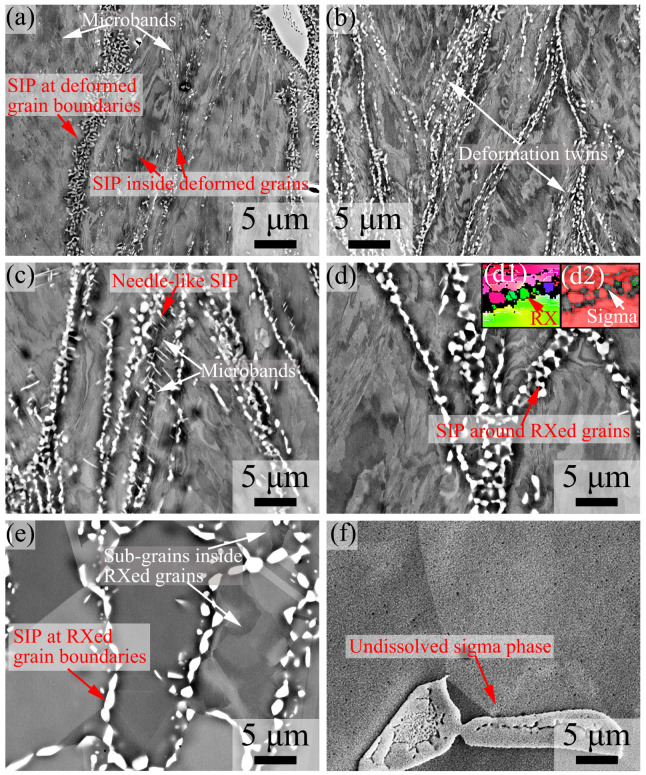
ECC images of 7Mo SASS after stress relaxation tests at 0.1 s^−1^, 50% strain, and 1000 s with different temperatures. (**a**) 750 °C; (**b**) 800 °C; (**c**) 850 °C; (**d**) 900 °C; (**e**) 1000 °C; (**f**) 1100 °C.

**Figure 7 materials-16-06401-f007:**
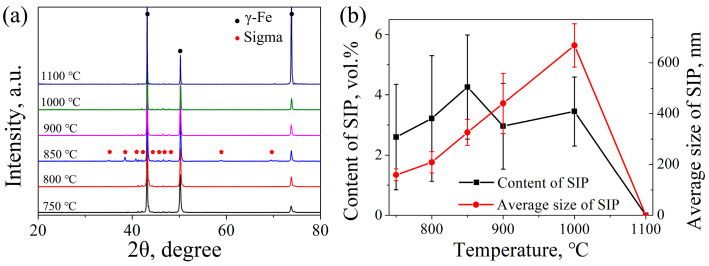
(**a**) XRD results and (**b**) the content and average size distribution of SIP of 7Mo SASS after stress relaxation test.

**Figure 8 materials-16-06401-f008:**
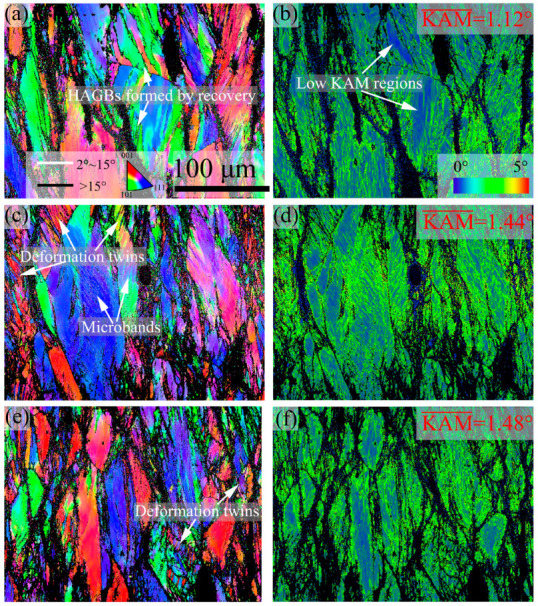
IPF and KAM images of 7Mo SASS after stress relaxation tests at 850 °C, 50% strain, and 1000 s with different strain rates. (**a**,**b**) 0.01 s^−1^; (**c**,**d**) 0.1 s^−1^; (**e**,**f**) 1 s^−1^.

**Figure 9 materials-16-06401-f009:**
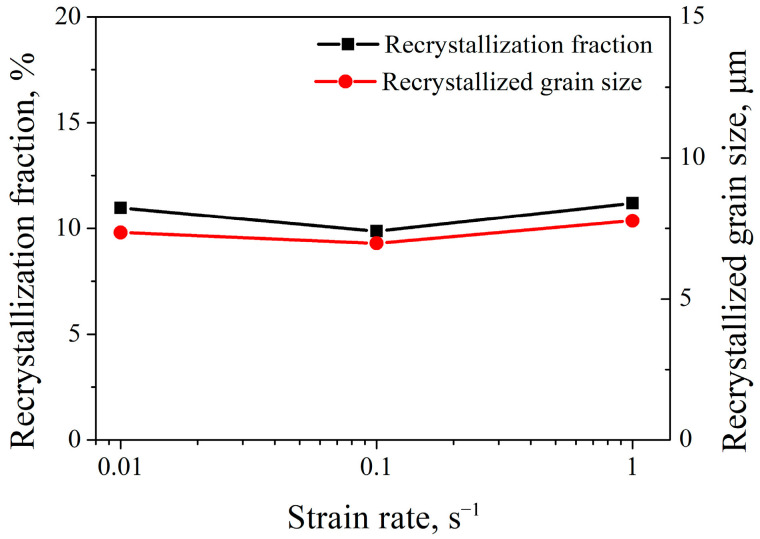
Recrystallization fraction and average recrystallized grain size of 7Mo SASS after stress relaxation tests under different strain rates.

**Figure 10 materials-16-06401-f010:**
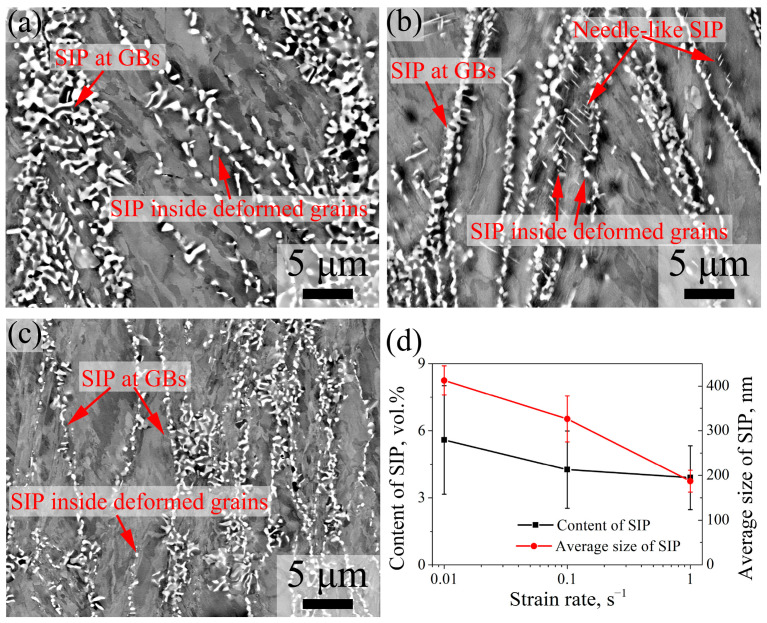
ECC images of 7Mo SASS after stress relaxation tests at 850 °C, 50% strain, and 1000 s with different temperatures. (**a**) 0.01 s^−1^; (**b**) 0.1 s^−1^; (**c**) 1 s^−1^; (**d**) statistical data.

**Figure 11 materials-16-06401-f011:**
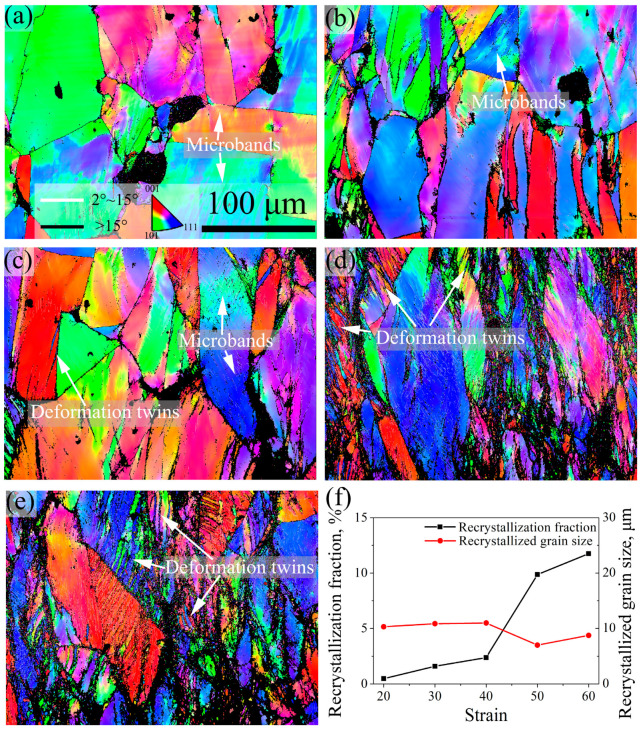
IPF images of 7Mo SASS after stress relaxation tests at 850 °C, 0.1 s^−1^, and 1000 s with different strains. (**a**) 20%; (**b**) 30%; (**c**) 40%; (**d**) 50%; (**e**) 60%; (**f**) statistical data.

**Figure 12 materials-16-06401-f012:**
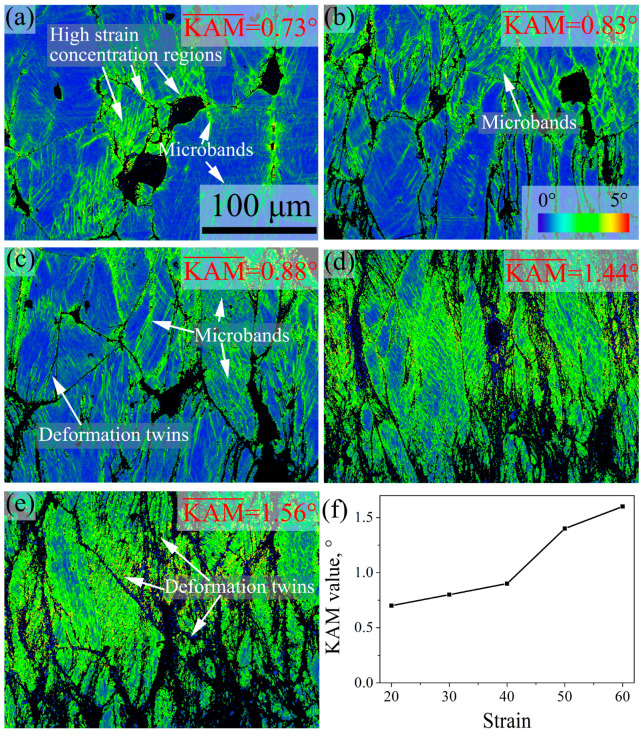
KAM images of 7Mo SASS after stress relaxation tests at 850 °C, 0.1 s^−1^, and 1000 s with different strains. (**a**) 20%; (**b**) 30%; (**c**) 40%; (**d**) 50%; (**e**) 60%; (**f**) statistical data.

**Figure 13 materials-16-06401-f013:**
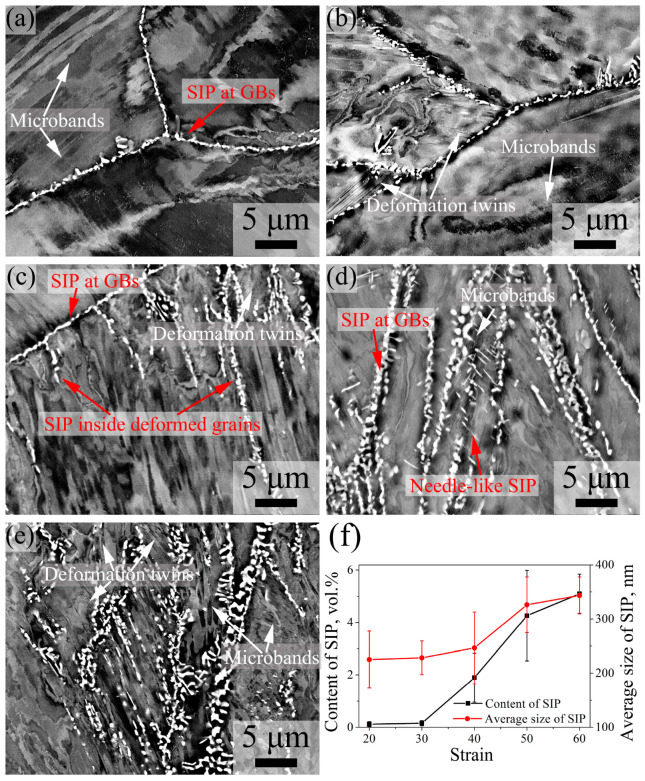
ECC images of 7Mo SASS after stress relaxation tests at 850 °C, 0.1 s^−1^, and 1000 s with different strains. (**a**) 20%; (**b**) 30%; (**c**) 40%; (**d**) 50%; (**e**) 60%; (**f**) statistical data.

**Figure 14 materials-16-06401-f014:**
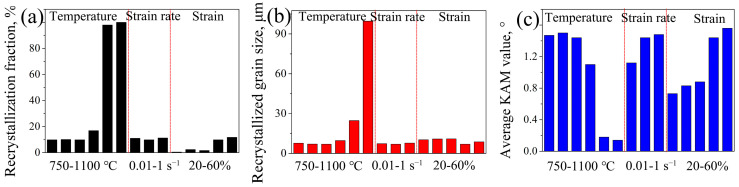
(**a**) Recrystallization fraction, (**b**) average recrystallized size, and (**c**) average KAM value of 7Mo SASS after stress relaxation tests under different conditions.

**Figure 15 materials-16-06401-f015:**
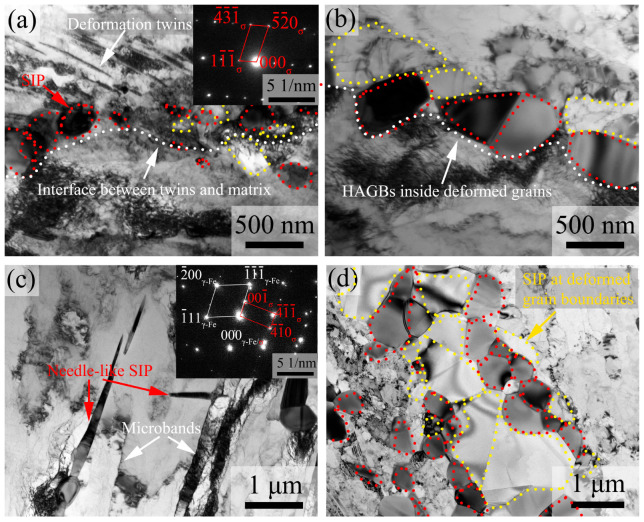
TEM images of SIP in 7Mo SASS after stress relaxation tests at 0.1 s^−1^, 50% strain, and 1000 s with (**a**) 750 °C and (**b**–**d**) 850 °C.

**Figure 16 materials-16-06401-f016:**
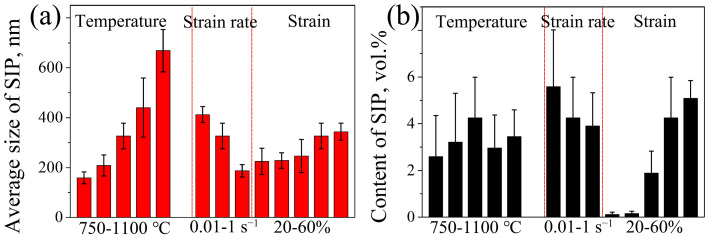
(**a**) Average size and (**b**) content of SIP in 7Mo SASS after stress relaxation tests under different conditions.

## Data Availability

Data are available upon request.

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
