# Peer review of "Effect of Deformation Conditions on Strain-Induced Precipitation of 7Mo Super-Austenitic Stainless Steel"

_materials, 2023, doi:10.3390/ma16196401_

Round 1
Reviewer 1 Report
Review Report #1
The authors presented an article titled “Effect of deformation parameters on the strain-induced precipitation behaviors of 7Mo super-austenitic stainless steel”. The "Materials" journal covers this article. However, the article will be ready for publication after a major revision. Comments are listed below. (The similarity rate is 24%.)
1. The introductory part is insufficient. It should be expanded.
2. The references given in the introduction are out of date. References should be increased by giving more current references.
3. What is the novelty of this article? The difference from previous studies in the literature should be explained.
4. An experimental scheme can be given in the material and method section.
5. Sample preparation should be explained in more detail.
6. There is no discussion in the Results sections. Similar studies in the literature should be compared and discussed.
7. In Figure 10, the resolution seems very low. Also, the texts are not readable.
8. Apparently, conclusions are just observations. The explanations given for the conclusions of the article need to be checked thoroughly.
9. The article contains numerous typographic and language errors. It should be corrected.
10. The article should be rearranged by taking into account the journal writing rules and citation rules.
Author Response
- The introductory part is insufficient. It should be expanded.
Authors’ response: We appreciate your suggestion and we have enhanced the Introduction Section accordingly. The specific content can be found in the highlighted red color regions.
- The references given in the introduction are out of date. References should be increased by giving more current references.
Authors’ response: We have updated the references in Introduction Section.
- What is the novelty of this article? The difference from previous studies in the literature should be explained.
Authors’ response: Previous studies of SIP behaviors were mainly focused on the carbide, nitride or carbonitride precipitates in microalloyed steels or nickel-based alloy, while few studies on the intermetallic precipitates in 7Mo SASS. Moreover, the extremely high content of alloying elements (especially Cr and Mo) of 7Mo SASS greatly facilitates the strain-induced intermetallic precipitation during hot deformation processes, such as sigma phase. For instance, Pu et al. observed a large number of strain-induced sigma phases with size range of 0.5~10 μm in 7Mo SASS after hot compression deformation at 1000 °C/ 0.001 s-1. The size of strain-induced sigma phases is significantly larger than the size of SIP (about 2~20 nm) in microalloyed steels or nickel-based alloy (Materials Science & Engineering A 598 (2014) 174–182). Due to the large content and size, the presence of these brittle intermetallic phases can deteriorate the hot working performance of 7Mo SASS, promote hot deformation cracking, and also deteriorate the corrosion resistance. Given the fact, we systematically studied the SIP behaviors of 7Mo SASS and further explained the influence of hot deformation parameters on the SIP behaviors. More detailed content is shown in the modified Introduction Section.
- An experimental scheme can be given in the material and method section.
Authors’ response: Thanks for the suggestion. We have given the schematic diagram of the stress relaxation tests in Figure 1(b).
- Sample preparation should be explained in more detail.
Authors’ response: Thanks for the suggestion. We have provided a more detailed description of the sample preparation process, as outlined in the highlighted red regions of the Materials and Methods Section. We also given the schematic diagram of sample preparation process in Figure 1(a).
- There is no discussion in the Results sections. Similar studies in the literature should be compared and discussed.
Authors’ response: After a carefully examination of the article, we have incorporated suitable discussions and implemented language revisions in Results Section. As shown in Line 196-206, we have detailed discussed the reasons for the significant difference on SIP size between 7Mo SASS in this study and microalloyed steels in other studies.
- In Figure 10, the resolution seems very low. Also, the texts are not readable.
Authors’ response: We have updated Figure 10. In addition, we refined the language after a carefully examination of the article.
- Apparently, conclusions are just observations. The explanations given for the conclusions of the article need to be checked thoroughly.
Authors’ response: After a carefully examination of the article, we have revised and expanded the conclusions based on the results and discussions.
- The article contains numerous typographic and language errors. It should be corrected.
Authors’ response: After a carefully examination of the article, we have incorporated suitable discussions and implemented language revisions.
- The article should be rearranged by taking into account the journal writing rules and citation rules.
Authors’ response: Thanks for the suggestion. We have rearranged the article according to the journal writing rules and citation rules.

Reviewer 2 Report
-
I believe that abbreviation SIP (strain-induced precipitation) is better to use only once when it appears for the first time, for instance, in line 12 (see lines 14, 151). What are abbreviations used for? Abbreviations are used to replace some complex expressions. However, respected authors use both abbreviations and expressions to be replaced (see lines 43, 52, 56, 58, 104, 151, 189, 231, 306, 378, 381).
English needs some improvements (for instance, increase in, not increase of (line 396)).
Author Response
I believe that abbreviation SIP (strain-induced precipitation) is better to use only once when it appears for the first time, for instance, in line 12 (see lines 14, 151). What are abbreviations used for? Abbreviations are used to replace some complex expressions. However, respected authors use both abbreviations and expressions to be replaced (see lines 43, 52, 56, 58, 104, 151, 189, 231, 306, 378, 381).
Authors’ response: We appreciate your suggestion and we have revised it by retaining only the initial occurrence and eliminating subsequent complex expressions. In addition, we have made appropriate language modifications.

Reviewer 3 Report
The authors of manuscript presents interesting results regarding strain-induced precipitation behaviors of 7Mo super-austenitic stainless steel under various deformation conditions studied by strain relaxation test.
However few details should be claryfied:
1-in the Introduction section, the signification of syntax SIP is not explicated.
2-in all figures with plots should be used line+different symbols (easy to be read on black-white listed paper.
3-what is the effect of quickly quenched in water on the microstructure? Could that induce a new mechanical-stress ?
4-the result presented in paragraph 4 are not compared with that from the literature.
5- too less references positions and very old also (only 2 from last 5 years)
Moderate editing of English language required
Author Response
1-in the Introduction section, the signification of syntax SIP is not explicated.
Authors’ response: Thanks for the suggestion. We have provided a detailed explanation of SIP, as shown in Line 42~47. Specifically, plastic deformation can induce a large number of lattice defects, such as dislocations and microbands, which can serve as nucleation sites and provide rapid diffusion channels for elements, thereby accelerating precipitation. This phenomenon is referred to as strain-induced precipitation (SIP).
2-in all figures with plots should be used line+different symbols (easy to be read on black-white listed paper.
Authors’ response: Thanks for the suggestion. We have revised all figures according to the suggestions.
3-what is the effect of quickly quenched in water on the microstructure? Could that induce a new mechanical-stress?
Authors’ response: The purpose of quickly quenched in water is to fix the microstructure following stress relaxation tests and prevent the formation of new second phases during slow cooling process. Due to the limited size of experimental sample (Φ 8*12 mm), the temperature variations across different regions during quenching are homogenized, thereby no thermal stress. The microstructure of experimental sample is primarily composed of austenite, and no phase transformation occurs during quenching process, thereby resulting in minimal structural stress. Therefore, quickly quenched in water will not induce additional mechanical stress.
4-the result presented in paragraph 4 are not compared with that from the literature.
Authors’ response: After a carefully examination of the article, we have incorporated suitable discussions in Results Section. As shown in Line 196-206, we have detailed discussed the reasons for the significant difference on SIP size between 7Mo SASS in this study and microalloyed steels in other studies.
5- too less references positions and very old also (only 2 from last 5 years)
Authors’ response: We have updated the references in Introduction Section.

Reviewer 4 Report
This paper deals with an interesting topic of Strain-induced precipitation behaviors of 7Mo super-austenitic stainless steel (SASS) under various deformation conditions. The authors conducted valuable experimental research. Overall, the analysis of the results provides valuable insights into the behavior of 7Mo SASS under different deformation conditions. It is well-supported by the presented figures and aligns with established principles in materials science and metallurgy. However, further quantitative data and statistical analysis might strengthen the conclusions drawn from these findings. In order to publish the paper, the methodology and results need to be restructured and improved. Suggestions are provided in the attached file.

Author Response
- The abstract needs to "summarize the work"; the authors should present the methodology employed in the study, highlight the main results obtained, and discuss the contribution of the work.
Authors’ response: We appreciate your suggestion and we have carefully revised the Abstract based on the suggestion.
- The introduction needs to be expanded, providing an overview of the current state of research, especially focusing on studies conducted in the last 5 years. In the introduction, less than 30% of the citations are from the last 10 years. If there are no studies in the same research area as the present work, the authors need to make this clear, emphasizing the innovation of their current research.
Authors’ response: Thanks for the suggestion. We have updated the references in Introduction Section.
- The materials and methods section need to be expanded, providing a sequential and more detailed description of all the processes carried out in the material characterization. It is also important to provide more detailed descriptions of how the tests were conducted to assess the results. It would be interesting to include a flowchart detailing all the laboratory procedures, and then the authors should provide detailed descriptions of the experimental procedures.
Authors’ response: Thanks for the suggestion. We have provided a more detailed descriptions of the specimen preparation process and experimental procedures, accompanied by corresponding schematic diagrams as depicted in Figure 1.
Figure 1. Schematic diagram of (a) specimen preparation process and (b) stress relaxation tests.
- The authors reference Figure 5 before Figure 4. Organize the numbering of the figures.
Authors’ response: Thanks for the suggestion. We have modified the corresponding text to avoid such mistakes.
- In lines 220-222, the authors state: "For 7Mo SASS in this paper, strain-induced sigma phases can occur in the early stage of deformation [16]." It is not clear whether this statement is based on the research results and corroborates with the findings inreference [16], or if this statement is a result of the work in reference [16]. There are other paragraphs in the text that have similar wording. The authors need to revise the text.
Authors’ response: Thanks for the suggestion. We have revised the text as follows: “Our previous research [26] reveals that for 7Mo SASS, strain-induced sigma phase can be observed at the initial stage of deformation (earlier than 20% strain).”
- The text is not easy to read. The paragraphs are lengthy, making it difficult to understand the discussions of the results. There are several acronyms that are not explained, such as IPF and KAM. The authors need to carefully revise the text.
Authors’ response: Thanks for the suggestion. We have modified the sentences in Results and Discussions Sections to enhance readability. Moreover, we have provided explanations for acronyms. For instance, IPF and KAM images represent inverse pole figure and kernel average misorientation images, respectively.
- The authors need to better explain the mechanism of particle-induced nucleation (PSN) in relation to SIP formation.
Authors’ response: The relationship between PSN and SIP have further explored in 4.2 Section. During deformation process, high-density dislocations, i.e large deformation stored energy, tend to accumulate around large SIP, thereby promoting recrystallization. This process is referred to as PSN. Subsequently, the boundaries of recrystallized grains can also serve as the nucleation sites for SIP, thereby promoting SIP. This process can be simplified as: . As the increase of holding time, a multi-layer structure of SIP/Recrystallization/SIP is gradually formed. Meanwhile, there is a continuous decrease of stored energy with increasing holding time. When the stored energy reaches a low level, recrystallization through PSN becomes difficult, thereby this process eventually ceases.
- What is the order of influence of deformation parameters on recrystallization fraction,and why is temperature considered the most influential?
Authors’ response: The order of deformation parameters on recrystallization fraction, from strong to weak, is as follow: temperature, strain and strain rate. The increase of temperature can significantly promote the migration of grain boundaries, thereby greatly promoting the occurrence of recrystallization. For instance, when the temperature increases from 900 °C to 1000 °C, the recrystallization fraction significantly increases from 16.9% to 97.9%. While, when the temperature is low (850 °C), the recrystallization fraction exhibits a limited sensitivity to both strain and strain rate. For instance, with the strain increases from 20% to 60%, the recrystallization fraction shows a small increase from 0.5% to 11.7%. The increase of strain results in a large increase of deformation stored energy, thereby promoting the nucleation of recrystallization. However, due to the lower temperature, the rate of recrystallization growth is low, thereby leading to a small increment of recrystallization fraction. Moreover, with the strain rate decreases from 1 s-1 to 0.01 s-1, the recrystallization fraction is low (about 10%) and remains almost unchanged. The decrease of strain rate provides long time for recrystallization, while also provides long time for recovery. The increase of recovery will consume more stored energy, thereby reducing recrystallization nucleation. Therefore, the recrystallization fraction remains unchanged.
- How does the decrease in strain rate impact stored energy, and why does it have negligible effects on recrystallization fraction?
Authors’ response: The decrease in strain rate results in an increase in deformation time, as well as recrystallization and recovery time. However, the recrystallization rate is slow due to low temperature (850 °C), leading to a limited consumption of deformation stored energy by recrystallization. Nevertheless, the degree of recovery continuously rises with increasing deformation time, thereby consuming more deformation stored energy. This consumption of deformation stored energy lowers the nucleation rate of recrystallization. Therefore, although the duration for recrystallization increases, there is almost no change in the fraction of recrystallization. In summary, the decrease of strain rate leads to an increase of recovery extent, consequently diminishing the stored energy. However, due to the decrease of stored energy and slow recrystallization rate at low temperature, there is a limited change in recrystallization fraction despite long recrystallization time.
- It’s important to introduce in the conclusion the main contribution of the paper, andnot only summarize the main results
Authors’ response: Thanks for the suggestion. We have appropriately modified the Conclusions Section based on the suggestion.

Round 2
Reviewer 1 Report
The authors completed the necessary revisions. This article may be accepted for publication in its final form.
Author Response
Thanks for your suggestion. We have modified the manuscript and resubmitted it to the system.